# Thermal Storage of Nitrate Salts as Phase Change Materials (PCMs)

**DOI:** 10.3390/ma14237223

**Published:** 2021-11-26

**Authors:** Marco A. Orozco, Karen Acurio, Francis Vásquez-Aza, Javier Martínez-Gómez, Andres Chico-Proano

**Affiliations:** 1Instituto de Investigación Geológico y Energético (IIGE), Quito EC170518, Ecuador; marco.orozco@geoenergia.gob.ec (M.A.O.); francisvasquezaza@gmail.com (F.V.-A.); 2Campus Miguel de Cervantes, Universidad Internacional SEK (UISEK) Ecuador, Calle Alberto Einstein s/n y 5ta. Transversal, Quito EC170134, Ecuador; karen.acurioc@gmail.com; 3Departamento de Teoría de la Señal y Comunicación, Área de Ingeniería Mecánica, Escuela Politécnica, Universidad de Alcalá (UAH), 28805 Alcalá de Henares, Madrid, Spain; 4Departamento de Ingeniería Química, Escuela Politécnica Nacional, Ladrón de Guevara E11-253, Quito EC170525, Ecuador; andres.chico@epn.edu.ec

**Keywords:** inorganic PCMs, thermal storage, nitrate salt mixtures

## Abstract

This study presents the energy storage potential of nitrate salts for specific applications in energy systems that use renewable resources. For this, the thermal, chemical, and morphological characterization of 11 samples of nitrate salts as phase change materials (PCM) was conducted. Specifically, sodium nitrate (NaNO_3_), sodium nitrite (NaNO_2_), and potassium nitrate (KNO_3_) were considered as base materials; and various binary and ternary mixtures were evaluated. For the evaluation of the materials, differential Fourier transform infrared spectroscopy (FTIR), scanning calorimetry (DSC), thermogravimetric analysis (TGA), and scanning electron microscopy (SEM) to identify the temperature and enthalpy of phase change, thermal stability, microstructure, and the identification of functional groups were applied. Among the relevant results, sodium nitrite presented the highest phase change enthalpy of 220.7 J/g, and the mixture of 50% NaNO_3_ and 50% NaNO_2_ presented an enthalpy of 185.6 J/g with a phase change start and end temperature of 228.4 and 238.6 °C, respectively. This result indicates that sodium nitrite mixtures allow the thermal storage capacity of PCMs to increase. In conclusion, these materials are suitable for medium and high-temperature thermal energy storage systems due to their thermal and chemical stability, and high thermal storage capacity.

## 1. Introduction

Efficient energy storage systems have emerged because of the interest in reducing the greenhouse gas (GHG) emissions caused by the increasing energy demand [1]. Consequently, it is essential to boost the current alternatives to optimize the use of available energy resources and minimize the environmental impact caused by the usage of fossil fuels. In this sense, the improvement of efficient energy storage systems represents a crucial option to accomplish these objectives [2]. Furthermore, it is well known that renewable sources of energy are intermittent. That is why it is crucial to ensure and provide stability by implementing energy storage systems [3].

Thermal energy storage (TES) systems can store energy generated in an environment and released it when required [2]. Heat storage can be achieved using sensible heat (SHS), latent heat (LHS), and thermochemical (TQS) storage [1]. Particularly, LHS presents superior characteristics for heat storage over the other ones because of its high latent heat and consistent phase change temperatures [4]. Correspondingly, the volume of the system is less affected by using latent heat storage systems over sensible heat systems due to their higher density and specific heat [5]. In this sense, LHS requires less volume for its function compared with SHS and TQS. Accordingly, phase change materials (PCM) present characteristics that make them an advantageous option for thermal energy storage.

In general, PCMs use the energy that a system absorbs or releases during its phase change (generally solid to liquid) and then dispose of that energy when it is needed [6,7]. Thermal storage by using PCMs is an opportunity to accomplish energy security with efficiency by taking care of the environment [8]. In addition, PCMs have characteristics such as high energy storage density and an isothermal process during the phase change that allow a small footprint, make them more attractive as TES systems, and have a competitive cost [9]. According to the operating temperature, there are different types of PCMs such as those with low, medium, and high latent heat thermal energy storage. In this way, the operating temperature is defined by the melting temperature of the PCM [10]. Moreover, for a PCM, it is important to establish its storage capacity by measuring the value of the heat of fusion. Additionally, other properties such as high enthalpy of fusion, the temperature of phase change within the operating range of the system, chemical and thermal stability, non-toxicity, non-flammability, availability, and affordability are imperative to select a PCM for a specific use [1]. Furthermore, phase change memory (PCM) devices are enabled by amorphization, and crystallization-induced changes in the devices’ electrical resistances [11].

On the other hand, it is possible to identify three types of PCMs based on their source: organic (fatty acids, paraffins, alkanes), inorganic (metals, alloys, salts, salt hydrates), and eutectics (organic and inorganic, organic-inorganic mixtures) [12]. Indeed, inorganic PCMs present advantages such as high energy storage. Inorganic PCMs have a higher capacity to store energy, higher operating temperatures, higher thermal conductivity, and lower cost compared with organic PCMs [12]. Even though inorganic PCMs have the drawback of being corrosive to metals, encapsulation of the PCM can be used to prevent this problem [13,14]. For example, Xu et al., (2017), analyzed the use of diatomite as a shape stabilization material for sodium nitrate. It was determined that the composite material presented low corrosiveness, high energy density, and high thermal stability [15]. In the same way, expanded vermiculite was used to form a composite with sodium nitrate. Similarly, good stability and low corrosiveness were achieved [4]. Additionally, salt hydrates present a disadvantage that is undercooling [6]. In this sense, it is better to work with dehydrated salts.

Particularly, in inorganic salts, the most used anions are nitrates, nitrate/nitrite mixtures, carbonates, chlorides, and fluorides, and cations belonging to alkaline elements such as sodium or potassium [3]. In this sense, materials such as potassium nitrate (KNO_3_), sodium nitrate (NaNO_3_), and sodium nitrite (NaNO_2_) have melting points at suitable temperatures to be used in thermal storage applications, exceeding 150 °C, such as solar thermal systems [5]. For example, a study by Bauer, T., determined, using DSC, the high enthalpy of fusion of NaNO_3_ (178 kJ·kg^−1^) [16]. Kourkova L. et al. evaluated the enthalpy of fusion of NaNO_2_. They found a temperature of fusion of 164 °C and enthalpy of fusion corresponding to 13.9 kJ·mol^−1^ [17]. Additionally, it was determined the melting temperature (336 °C) and the high enthalpy of fusion of KNO_3_ (116 kJ·kg^−1^) by calorimetry [18].

Furthermore, studies have been developed to enhance heat transfer and energy storage applications with nano/microparticles suspended [19,20,21,22,23,24]. Some examples can be seen in the research of Ho et al., Forced convection heat transfer of Nano-Encapsulated Phase Change Material (NEPCM) suspension in a mini-channel heatsink [20], in which NEPCMs with particle sizes in the range of 250–350 nm were synthesized. The synthesized NEPCM–water suspension is employed as the working-fluid for heat removal from a microchannel heatsink. Further research that shows the effect of microencapsulation is that developed by Eisapour et al., Exergy and energy analysis of wavy tubes photovoltaic-thermal systems using microencapsulated PCM nano-slurry coolant fluid [21], in which a more efficient water-cooled photovoltaic-thermal system with wavy tubes was developed. Additionally, the consequences of coolant fluid including water, Ag/water nanofluid, microencapsulated phase change material slurry, and a new type of cooling fluid called microencapsulated phase change material nano-slurry, were studied. Hajjar et al., in Time periodic natural convection heat transfer in a nano-encapsulated phase-change suspension [22], developed the natural convection of NEPCMs suspension in a cavity with a hot wall with a time-periodic temperature. In addition, the article of Talebizadehsardari et al., Consecutive charging and discharging of a PCM-based plate heat exchanger with zigzag configuration [23], studied the remarkable energy savings, isothermal nature of the operation and low costs, energy storage with a plate type heat exchanger with zigzag configuration.

In this way, SiO_2_/Al_2_O_3_ nanoparticles added to Solar Salt can increase the heat of fusion by 7.4% and the specific heat by 52.1% in the solid phase and 18.6% in the liquid phase [19]. Another study found that by adding 1.0 wt % of silica as nanofluid into Solar Salt, it is possible to increase the heat capacity by 26.7% [25]. Additionally, CuO-doped nitrate salts were studied, evidencing increased thermal conductivity, diffusivity, and stability [26]. Nevertheless, a negative effect has been determined with the use of nanoparticles into the salts for the abovementioned purpose due to a notable increase in the corrosiveness of the salts [27,28,29,30,31,32].

Moreover, binary and ternary combinations of these salts have shown improved properties. To illustrate, the work developed by Berg [6] presents the binary phase diagram between NaNO_2_ and NaNO_3_. This study determined that the mixture is a e.

Eutectic system with a solid point at 230 °C between a range of 0.25 to 0.80 of NaNO_3_ molar fraction. Additionally, the combination of NaNO_3_ and KNO_3_ in percentages of 60 and 40, respectively, is a mixture is known as “Solar Salt”, whose eutectic is around 54% KNO_3_ and 46% NaNO_3_ and its melting point is approximately 222 °C, its minimum operating temperature is 290 °C, and the maximum operating temperature is about 585 °C [7,8,9,10,11,12,16,17,18,27,28,29,30,31,32,33,34,35,36,37,38,39,40,41,42,43,44,45,46,47,48,49,50,51,52,53,54,55,56,57,58,59,60,61,62,63,64,65,66,67,68,69,70]. Previous studies indicate that the mixture of these two compounds, in the indicated proportions, presents advantages such as high energy density, high heat capacity, low cost (about 6.68 $/kg), and suitable properties for their operation at high temperatures [7,8,9,10,11,12,16,17,18,27,28,29,30,31,32,33,34,35,36,37,38,39,40,41,42,43,44,45,46,47,48,49,50,51,52,53,54,55,56,57,58,59,60,61,62,63,64,65,66,67,68,69,70,71,72]. Additionally, according to [37] and [36], the maximum temperature or the limit of stability of the samples is normally defined as the temperature at which the sample has lost 3% of its initial weight. Moreover, it was stablished that the NaNO_3_ initial degradation temperature, 400 °C, is lower than the decomposition temperature of the material, 450 °C, showing thermal stability [10]. On the other hand, NaNO_2_ degrades above 330 °C. In this way, the first isothermal process must be reduced to 300 °C, ensuring that there is no weight loss and avoiding thermal instability [11]. Finally, KNO_3_ loses 0.044% of its weight at 180 °C, presenting stability [12]. These studies show the suitability of using NaNO_3_, NaNO_2_, and KNO_3_ as thermal storage media due to their thermal stability. Nonetheless, it has been determined that eutectic mixtures of them are more promising for TES, as shown in [12].

Most of the current literature on inorganic salts as PCMs pays particular attention to binary or ternary mixtures because they provide advantages such as low costs, low vapor pressures, and high thermal stability [69]. In this way, it is essential to note the difference between eutectic and non-eutectic mixtures. Eutectic mixtures allow lower melting temperatures than the pure components, and at the same time, there is no segregation during the melting process [12]. On the other hand, although non-eutectic mixtures present segregation, they provide a higher range of temperature of phase change. Hence, mixtures of NaNO_3_, NaNO_2_, and KNO_3_ have been identified as non-eutectic mixtures [7,51,52,53]. Nonetheless, further study is required due to the discrepancies. There is a contradiction in the previous analyzes carried out by different researchers; some determine that this combination is eutectic, and others determine that this system is of the continuous solid solution type [6].

Many studies have been focused on the use of nitrate salts as phase change materials. It has been demonstrated that the use of nitrate salts as PCMs have several applications such as small housing solar plants, water heating systems, solar cookers, and solar dryers [56,57,58,59,60,61]. To illustrate, a solar box cooker was fabricated with a ternary mixture of NaNO_3_, NaNO_2_, and KNO_3_ as PCM. It was found that the PCM allows storing energy that can be used when solar radiation is absent. To be specific, approximately 108% higher was the load cooking time when using the PCM [58,59,60,61,62,63,64,65]. Moreover, a solar organic Rankine cycle plant was tested with the use of the non-eutectic mixture of NaNO_3_, NaNO_2_, and KNO_3_ as the thermal storage system in a solar plant of 100 KWh [66,67]. Additionally, Prieto and Cabeza (2019) have used a cascade arrangement with different salts as PCMs including NaNO_3_ that allowed a lower cost and higher energy storage in solar power plants [66].

In the same token, concentrated solar power plants take advantage of the large thermal stability range of nitrate salts and use them as thermal storage systems [68]. In this sense, several studies have focused their attention on analyzing the thermal and chemical properties of these salts. Particularly, the effect of nitrogen and oxygen atmospheres on the performance of Solar Salt, the mixture of NaNO_3_ and KNO_3_, and Hitec a ternary mixture of 53 wt % KNO_3_, 7 wt %, NaNO_3_, and 40 wt % NaNO_2_ was determined [24]. The former was stable under the mentioned atmospheres [67]. Nonetheless, Hitec presents the disadvantage that NaNO_2_ oxidizes under the presence of oxygen. Hence, it was established that inert atmosphere is required above 300 °C with nitrogen as an inert gas [70,71].

In this context, this research work determines the thermo-physical properties of nitrate salts (NaNO_3_, NaNO_2_, and KNO_3_) and their binary and ternary combinations due to the discrepancy and errors associated with different experimental setups conducted in different studies. In this sense, this study tries to overcome the limitations in previous literature. Specifically, the properties of different salt mixtures are evaluated under the same methodological conditions. Based on this, nitrates as PMC were characterized through Fourier-transform infrared spectroscopy (FT-IR), differential scanning calorimetry (DSC), thermogravimetric analysis (TGA), and scanning electron microscopy (SEM) to establish their feasibility as PCMs for renewable energy storage applications.

## 2. Materials and Equipment

This section presents the selected materials and techniques used to evaluate the thermal, chemical, and morphological properties of three nitrate salts: sodium nitrate (NaNO_3_), potassium nitrate (KNO_3_), and sodium nitrite (NaNO_2_). The characterization was developed for the pure components and various combinations in different percentages by mass. For this research, the materials used were KNO_3_ of LOBA Chemie with 99% purity [73], NaNO_2_ of LOBA Chemie with 99% purity [74] and NaNO_3_ of LOBA Chemie with 99% of purity [75].

The following assay techniques were used: FTIR, to identify the material used; DSC, to find the enthalpy of phase change and specific heat; THM, as a complementary method to DSC to test the sample in a larger volume; TGA, to establish the thermal stability of the compound; and finally SEM, to verify the morphological characteristics of the sample after the heating process.

### 2.1. Materials

The selected granulated materials are KNO_3_, NaNO_3_, and NaNO_2_, which will be arranged in binary and ternary mixtures. To define these combinations, the phase diagram was considered, which shows the thermal behavior of the mixtures. Table 1 shows the name established for each compound, the percentage by mass of each combination, and the performed tests. Particularly, for binary compounds, the work developed by Berg [6] was consulted, which presents the binary phase diagram between NaNO_2_ and NaNO_3_, considering the results of DSC tests and Raman spectroscopy. For the selection of the percentages of the binary compound mixture, the eutectic affinity of the materials involved and the need to consider a certain operating temperature for energy storage were considered. Additionally, the combination of NaNO_3_ and KNO_3_ in percentages of 60 and 40, respectively, was taken as a reference. This mixture is known as “Solar Salt” and has been extensively investigated. 

It is important to mention that the analyzed percentages were selected around the eutectic point of the mixture to verify whether the variations in the results found are significant. In Table 2, the thermal properties of the most common compounds found in the literature are shown. To verify that a PCM is suitable and meets the above selection criteria, it is important to determine its thermophysical properties. Conventional thermal characterization techniques include DSC analysis and differential thermal analysis (DTA). The high cost of the test equipment, its complexity, the lack of visualization of the phase change in the material, and the use of small samples of material cause the results found to vary because sometimes the behavior of a PCM depends on the quantity of it [33].

### 2.2. FTIR Analysis

The thickness of the sample must be appropriate for reducing the radiant power of the detector at the absorption frequencies used in the analysis. Before starting the procedure, preliminary preparation of the sample is important. In this case, due to the inherent hygroscopicity of molten salts, a drying pre-treatment is necessary. The procedure was verified by periodically measuring the variation in the weight of the samples every hour. The material containing the compound must be transparent to infrared radiation, such as potassium bromide (KBr), to ensure the measurement of absorption only of the sample. For solid samples, a small amount of pulverized KBr was mixed in an agate mortar and made into a pellet. The equipment used for the FTIR analysis was Perkin Elmer, Frontier model (Quito, Ecuador). The analysis was conducted in the range of 350–4000 cm^−1^. This test was carried out for the three pure salts, according to Table 1 compounds 1, 2, and 3.

### 2.3. Differential Scanning Calorimetry (DSC)

To perform these analyses, a Metler Toledo differential scanning calorimeter was used. An amount of 5 to 7 mg of the samples were collocated in a pan. The heating rate was 10 °C/min [44]. For the analysis of the results, the thermal analysis system “STARe Evaluation Software” developed by the Mettler Toledo company (Quito, Ecuador) was employed.

According to Table 1, all compounds were tested by this method. The DSC analysis was developed in a Meter Toledo calorimeter, model: HP DSC 1, which consists of a chamber refrigeration equipment. Materials such as Indium and Zinc were used as reference elements to calibrate the equipment. According to [2], the phase change enthalpy was obtained according to Equation (1).
(1)$$\Delta H=\int_{t_{1}}^{t_{2}}\phi dt=\int_{t_{1}}^{t_{2}}\frac{dH}{dt}dt$$

For the calculation of the *C_p_*, Equation (1) is derived as a function of time. Where *C_p_* is the specific heat of the sample, Ø is the heat flux; $\beta_{S}$ is the heating rate (dT/dt) and m is the mass of the compound.
(2)$$C_{p}=\frac{dH/dt}{dT/dt}\frac{1}{m}=\frac{\emptyset }{\beta_{s}m}$$

### 2.4. Termogravimetry Analysis (TGA)

A thermogravimetric analyzer SHIMADZU, TGA50 (Quito, Ecuador) was used. An amount of 2 to 4 mg of the samples were evaluated between 25 and 600 °C. A heating rate of 10 °C/min under an inert atmosphere of N_2_ was used. This test has been carried out on compounds 1, 2, and 3 (see Table 1).

### 2.5. Scanning Electron Microscopy (SEM)

For this analysis, an electron microscope brand was used: TESCAN VEGA 3 SEM (Quito, Ecuador). Resolution in high vacuum mode ranges from 2 nm to 30 keV, and in low vacuum mode from 3.5 nm to 30 keV. The material is introduced into the sample holder together with a carbon filament to improve conductivity. As it is a non-conductive material, the surface must be metalized with gold under vacuum, applying a time of 60 s in the process. The parameters, acceleration voltage (HV), and working distance (WD) are important since they are related to the resolution and contrast of the image. This test has been carried out on compounds 1, 2, and 3 (see Table 1).

## 3. Results and Discussion

KNO_3_, NaNO_2_, and NaNO_3_, nitrate salts, and their binary and ternary mixtures were chemically and thermally characterized. Particularly, FTIR, DSC, TGA, and SEM analyses of the samples were conducted to obtain conclusive data about the workability of the nitrate salts as PCMs and avoid errors associated with the experimental setup.

### 3.1. FTIR Analysis

Through the FTIR analysis, it is possible to identify the functional groups of the molecules that make up the analyzed material, meaning that the composition of the sample ould be known. The technique uses the infrared region of the electromagnetic spectrum.

This section shows the results of the characterization of sodium nitrate, potassium nitrate, and sodium nitrite according to the FTIR technique. Figure 1 shows several significant peaks such as the peak with the highest absorbance, 0.65, and occurs at a frequency of 1271.55 cm^−1^, corresponding to sodium nitrite. The analysis of the NaNO_3_ compound in Figure 2 shows a peak with the highest absorbance, with a value of 4.89, at a frequency of 1370.02 cm^−1^ corresponding to sodium nitrate. Likewise, the analysis of KNO_3_ presents the peak with the highest absorbance of 0.85 with a frequency of 1384 cm^−1^ in Figure 3. For the case of NaNO_3_ and KNO_3_, the quantitative study of absorbance is around the band at 1385 cm^−1^ (present in the IR spectra of the three compounds in KBr medium). Therefore, it can be stated that there is no solid dissolution between KBr and the compounds analyzed [48].

Specifically, the presence of the peak at 1385 cm^−1^ is attributed to antisymmetric stretching of N–O vibration. In addition, in the presence of a peak at 825 cm^−1^, characteristics of the N–O out-of-plane bending, can be observed [26]. The abovementioned peaks confirm that the analyzed materials correspond to nitrate salts. Nonetheless, there was evidence for the presence of the O–H stretching around 3440 cm^−1^. Although the samples were dehydrated, this peak shows water absorption [54]. Moreover, in the case of the analysis of inorganic compounds, other characteristic values of absorbance and wave number can be observed. These values are captured due to the presence of impurities in the compound.

### 3.2. Differential Scanning Calorimetry (DSC)

The results of the DSC analysis allowed variables to be determined such as enthalpy of fusion, starting and ending temperatures of the endothermic process, and specific heat.

#### 3.2.1. Enthalpy of Fusion and Melting Temperature

Table 3 shows the evaluation of the compounds by DSC. Additionally, the DSC curves of compounds 1, 2, and 3 (KNO_3_, NaNO_3_, and NaNO_2_) are presented in Figure 4.

KNO_3_ compound exhibits two endothermic processes. The first corresponds to the solid–solid transition at a temperature of 130 to 135 °C with an enthalpy value of 47 kJ/kg. As the temperature increases, another endothermic process is generated at 333 to 338 °C. This phase change corresponds to the solid to liquid (S–L) transition, since the starting temperature of the process and the melting point occur at the temperature peak with the highest heat flux, that is, at 334 °C, and the enthalpy of fusion reaches a value of 97 kJ/kg. The last value approximately doubles the value found during the solid to solid (S–S) transition process. The melting temperature around 333 °C matches an early study developed by Mohamed et al. [12]. Nevertheless, the heat of fusion was determined as 266 kJ/kg. This value is 85% higher than the value established in this study. Hence, this difference could be attributed to the impurities of the sample, as the FT-IR analysis showed.

Moreover, for NaNO_3_, the first endothermic process begins at a temperature of 271 °C and ends at 277 °C, presenting a tendency change at 276 °C. This is correlated to the S–S transition. In the same way, it was observed that the endothermic process produces an increase in the heat flux, generating a change in enthalpy of 107 kJ/kg. Subsequently, the increase in temperature generates the S–L transition process from 305 to 311 °C. Specifically, the phase change occurs at 306 °C with 166 kJ/kg as the value of the enthalpy of fusion. According to Li et al., the enthalpy of fusion and the melting temperature of the sodium nitrate are 178.6 kJ/kg and 306.4 °C, respectively [4]. Consequently, the findings of the current study are consistent with the abovementioned ones.

Furthermore, the analysis of NaNO_2_ presents similar characteristics to Compound 2 (NaNO_3_) with the difference that the first endothermic process begins at 163 °C with a peak temperature at 165 °C. The process ends at 167 °C, and presents a change in enthalpy of 11 kJ/kg. The transition (S–L) occurs at 281 °C with an enthalpy of fusion of 221 kJ/kg. Kourkova et al. presented the thermal properties of NaNO_2_ of 280 °C as the melting point and 221.7 kJ/kg as the latent heat of fusion [17]. Therefore, this study agrees with previous findings. It can be seen from figure x that NaNO_3_ has a better performance than PCM due to its higher enthalpy of fusion compared with KNO_3_ and NaNO_2_.

Compound 4, 50% of NaNO_3_ and 50% of NaNO_2_, is a binary mixture that exhibits two transition processes, one from 163 to 167 °C. A peak has been obtained at the temperature of 165 °C which indicates the transition temperature S–S. Likewise, similar values are observed for the second endothermic process, from 228 to 239 °C. The phase change temperature is 231 °C and its phase change enthalpy is 186 kJ/kg.

In the same way, compound 5, 53.5% of KNO_3_ and 46.5% of NaNO_3_ (solar salt) present an endothermic process from 107 to 120 °C. The existence of a peak is observed at the temperature of 111 °C. The solid–solid transition corresponds to the first endothermic process. The second endothermic process presents a perceptible steeply sloping peak, whose heat flux is approximately eight times greater than the first process. This last process represents the change in state from solid to liquid, which begins at 221 °C and ends at 226 °C with a temperature peak at 223 °C. The analyzed mixture was determined to have a melting temperature of 223 °C, an enthalpy of melt of 92 kJ/kg. The present findings seem to be consistent with other research which found that the enthalpy of fusion of this mixture is 106 kJ/kg and the melting temperature is 223 °C [12]. The slight difference in the values of the enthalpy of fusion can be explained in part by the presence of impurities in the samples.

For the compound 6, which is 46.5% KNO_3_ and 53.5% NaNO_3_, the test shows that as temperature increases, two endothermic processes occur, the first one from 107 to 118 °C. This process is quantified by its solid–solid transition enthalpy (S–S) of 23 kJ/kg. The second process was from 222 to 235 °C. In this process, the phase change (S–L) occurs, whose enthalpy of fusion is 114 kJ/kg. It is important to note that the peak value of this second process indicates the temperature at which the material changes from a solid to a liquid state; in this case at 225 °C. It is important to note that by switching the composition of the mixture, compounds 5 and 6, a higher enthalpy of fusion is achieved. This result is desired due to the increase in energy storage required in a PCM. In addition, compound 7 exhibits similar thermal properties to compound 6.

On the other hand, compounds 8, 9, and 11 are ternary mixtures. In Table 3, it is possible to see that compound 11 presented a better performance as PCM due to its higher enthalpy of fusion. From the evaluated compounds, although the selection of a PCM depends on the required temperature of phase change, it can be concluded that NaNO_2_ will have a better performance because of its higher energy storage capacity. Between the mixtures, the performance of 50% of NaNO_3_ and 50% of NaNO_2_ (compound 4), stands out compared to the other mixtures.

#### 3.2.2. Specific Heat

It is important to mention that is not possible to determine the specific heat during endothermic processes. Based on the DSC analysis results, this parameter was calculated by using methodology in [39], which relates the heat flow, the heating rate, and the mass of the compound. Table 4 shows the results of the specific heat for the solid and liquid state of the compounds, in the intervals where sensible heat exists.

In general, the compounds present specific heat values higher than 1000 J/kg·K either in a solid or liquid state, with exception of compounds 1 and 2 which show lower values. Compound 8 has the highest specific heat in solid and liquid states of 2597 J/kg K and 7611.43 J/kg K, respectively. Compound 3 has the second highest specific heat values of these compounds, which are 1848.42 and 1600.00 J/kg·K in the solid and liquid state, respectively. The lowest values were found to correspond to compound 1, which has values of 507.69 J/kg·K in solid-state and 739.13 J/kg·K in a liquid state. In the literature, it was found that the Cp in liquid state for compound 1 was 1190 J/kg·K, 1560 J/kg·K for compound 2, and 1350 for compound 5 [38]. Although these results differ from some published studies [42], the evidenced differences can be explained for the different methods used to determine the *C_p_*. To be specific, the methodology applied in this study was numerical, and on the studies used for the comparison, there was an experimental *C_p_* calculation based on a calorimeter.

### 3.3. Termogravimetry Analysis (TGA)

One of the important parameters to select a PCM is to evaluate its operating temperature. Thermogravimetry analysis will help us determine the material’s thermal stability; that is, it will find the variation in mass that the sample would experience as a function of the increase in temperature, which is caused by the fact that the system is not a hermetic system. This result made it possible to technically argue the feasibility of using PCM in low, medium, and high temperature thermal storage applications.

When exposed to high temperatures, alkali metal nitrate salts thermally decompose according to three different mechanisms [45]: (a) nitrite formation in the melt and oxygen release, (b) alkali metal oxide formation in the melt and nitrogen or nitrogen oxides release, and (c) vaporization of the nitrate salts.

Table 5 shows the results of weight variation in the samples analyzed by the TGA method. According to Table 5, point T_3_, the maximum temperature or the stability limit where the sample has lost 3% of its initial weight [46,47], is exceeded during the test carried out on NaNO_3_, even reaching 5.25% loss in weight at 600 °C. The same case is observed for KNO_3_, which presents a loss of weight of 4.41% at 600 °C. It was found in the literature that the degradation temperature of NaNO_3_ sample starts at 400 °C [48]. Nonetheless, this value is below the temperature of decomposition of the material, 450 °C. On the other hand, to avoid the instability of NaNO_2_, the first isothermal process was decreased to 300 °C [45].

In Figure 5, the results obtained from the TGA analysis of the three compounds are shown. Two curves stand ou; the first of them (cyan color), constitutes the reference of the initial mass of the sample, while the second (red color) shows the variation in this parameter as a function of the increase in temperature. The results obtained agree with the research carried out by [45]. The weight variation in the heating stage up to approximately 600 °C does not present a pronounced slope as observed in the case of bringing the material up to a temperature of 1000 °C. In this sense, these results show that the analyzed materials are thermally stable below 550 °C. Therefore, those materials can work stably as PCMs behind 600 °C.

### 3.4. Scanning Electron Microscopy (SEM)

This section presents the results obtained from the pure salts KNO_3_, NaNO_2_, and NaNO_3_ using the scanning electron microscope (SEM). In Figure 6, four KNO_3_ images are shown with different image magnifications. The same acceleration voltage (HV) of 20.0 kV has been maintained for all observations. The working distance (WD) is in the range of 13.49 to 14.40 mm. With these parameters, the images with the best contrast were obtained. Grains can be seen with a size that varies from approximately 0.1 to 0.5 mm and no porosity or cracks are observed. Irregularly shaped particles are observed that are dispersed over the entire surface, agglomerating some particles with others.

Four images of NaNO_3_ are shown in Figure 7. An acceleration voltage (HV) of 5.0 kV has been maintained, except for the last image. The working distance (WD) is in the range of 9.73 to 13.95 mm. No porosity or cracks are observed, and 250 to 500 µm of grains can be observed. Irregularly shaped particles are observed that are dispersed over the entire surface, agglomerating some particles with others. At higher magnifications of the microscope, flat surfaces are observed that will improve the reaction between particles.

In Figure 8, four images of compound NaNO_2_ are shown. An acceleration voltage (HV) of 5.0 kV has been maintained. The working distance (WD) is around 10 mm. No porosity or cracks are observed and 250 to 500 µm grains can be observed. Irregularly shaped particles are observed that are dispersed over the entire surface, agglomerating some particles with others.

## 4. Conclusions

In this study, the energy storage potential of nitrate salts and their mixtures for different purposes in energy systems from sustainable and renewable resources was evaluated. The FTIR analysis of the base salts confirmed the purity of the compounds. However, it was found that KNO_3_ presents water in its content. Moreover, from the TGA results, the NaNO_3_ and KNO_3_ materials exceed the limit temperature of degradation. This characteristic makes these materials not applicable for high-temperature energy storage applications.

On the other hand, emphasis is particularly placed on the greater enthalpy of fusion observed in sodium nitrite, 220.72 J/g. Nevertheless, it is important to consider not only this parameter but also the operating temperature of the system. Consequently, this PCM without mixtures will not be suitable for systems with operating ranges less than 278 °C.

In addition, the mixture of 50% NaNO_3_ and 50% NaNO_2_ presented an enthalpy of 185.6 J/g with a phase change start and end temperature of 228.4 and 238.6 °C, respectively. This result indicates that mixtures with sodium nitrite allow the thermal storage capacity of PCMs to increase. In this sense, it has been determined that this binary mixture will perform as a good PCM for medium-temperature storage systems.

In general, these materials are suitable for medium and high-temperature thermal energy storage systems due to their thermal and chemical stability and their high thermal storage capacity. In this sense, the results presented in this study can be used as an available source of thermal parameters of nitrites and nitrates as phase change materials obtained under a validated methodology. Therefore, these outcomes can be useful for the application of these PCMs in thermal energy storage systems.

## Figures and Tables

**Figure 1 materials-14-07223-f001:**
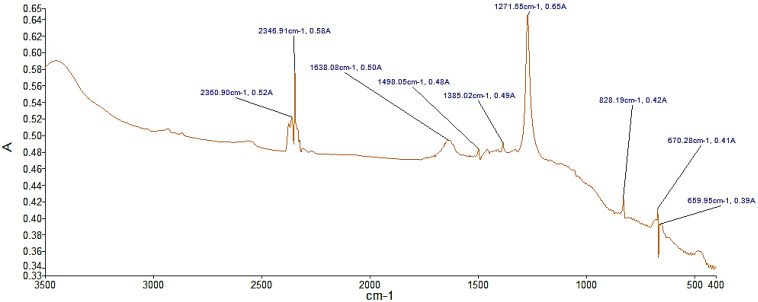
FTIR analysis of the NaNO_2_ sample. A indicates absorbance.

**Figure 2 materials-14-07223-f002:**
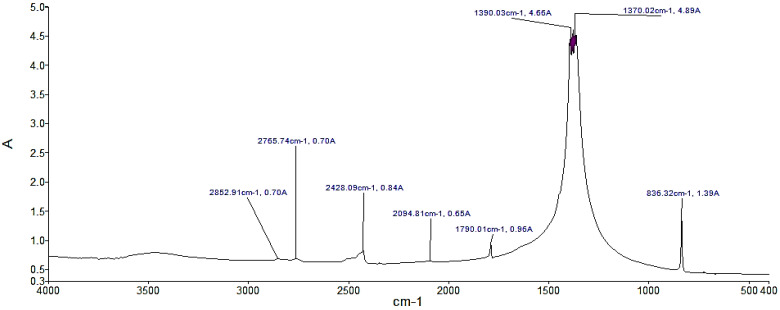
FTIR analysis of the NaNO_3_ sample. A indicates absorbance.

**Figure 3 materials-14-07223-f003:**
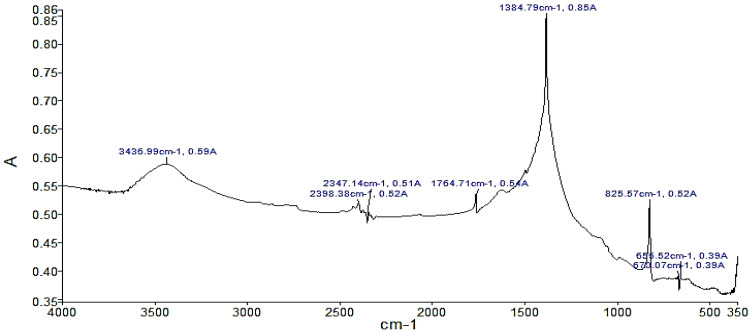
FTIR analysis of the KNO_3_ sample. A indicates absorbance.

**Figure 4 materials-14-07223-f004:**
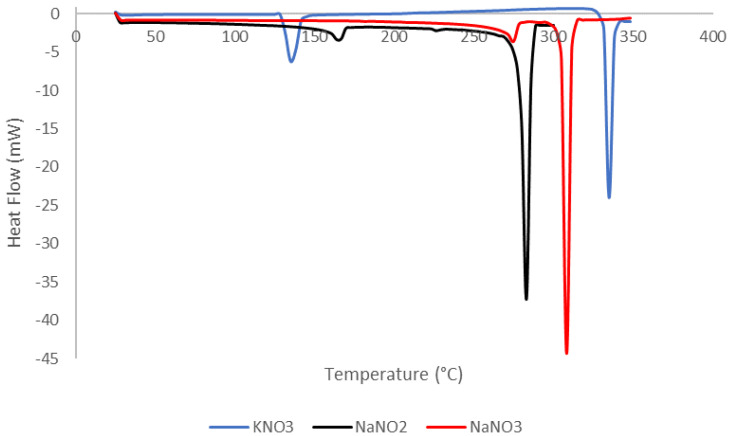
DSC curves of KNO_3_, NaNO_2_, and NaNO_3_.

**Figure 5 materials-14-07223-f005:**
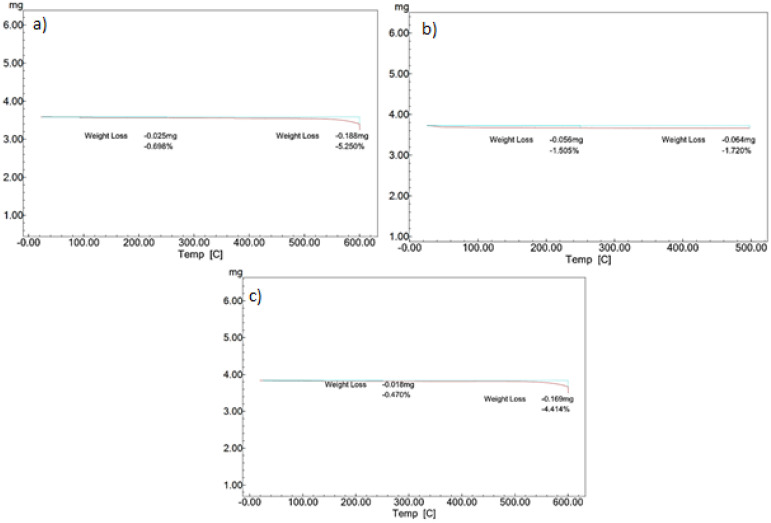
TGA analysis of (**a**) KNO_3_. (**b**) NaNO_3_. and (**c**) NaNO_2_.

**Figure 6 materials-14-07223-f006:**
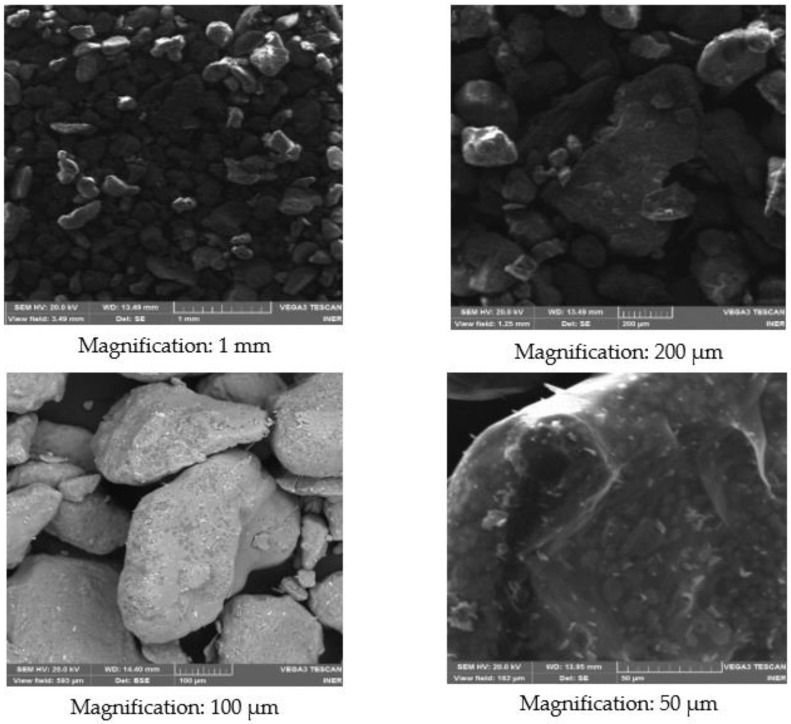
SEM images of the KNO_3_ sample.

**Figure 7 materials-14-07223-f007:**
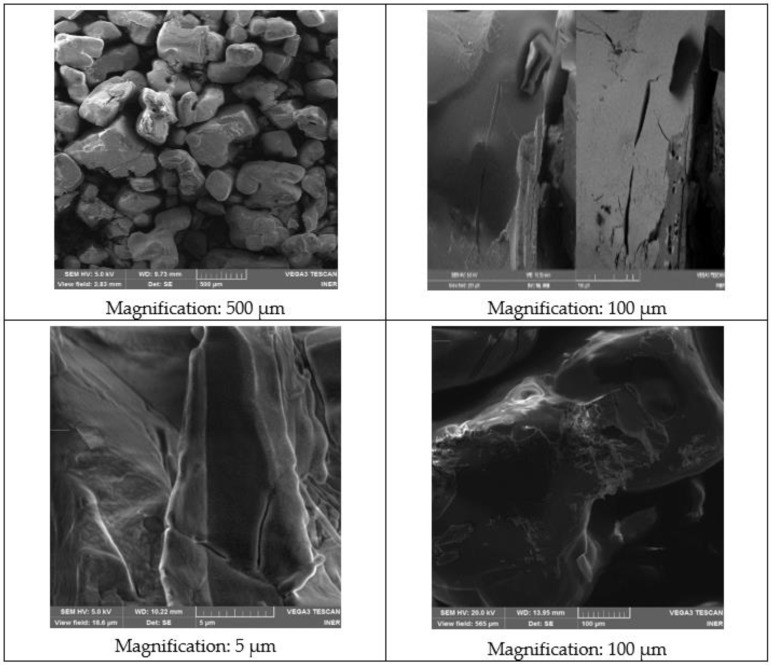
SEM images of the NaNO_3_ sample.

**Figure 8 materials-14-07223-f008:**
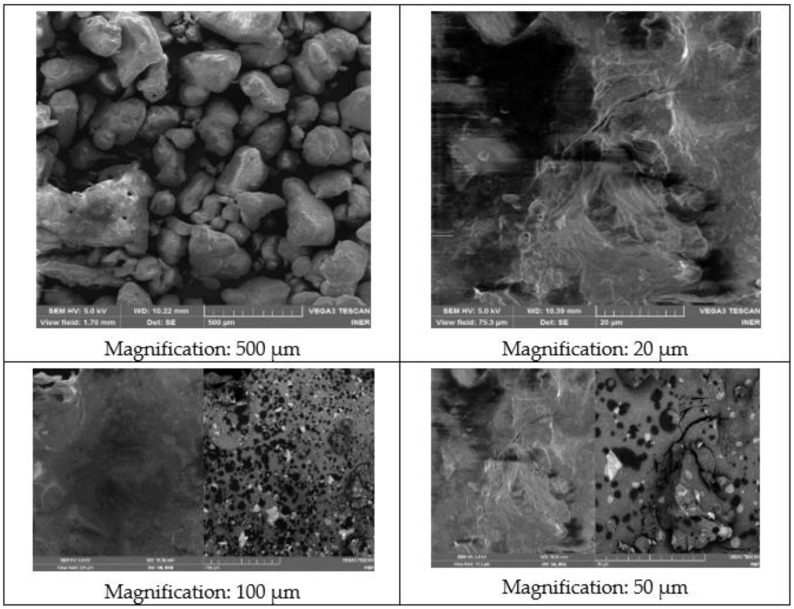
SEM images of the NaNO_2_ sample.

**Table 1 materials-14-07223-t001:** List of compounds.

Sample Number	Mass Percentage of the Mixtures			Assay Techniques				
	KNO_3_ (wt %)	NaNO_3_ (wt %)	NaNO_2_ (wt %)	FTIR	DSC	THM	TGA	SEM
1	100	0	0	X	X		X	X
2	0	100	0	X	X		X	X
3	0	0	100	X	X		X	X
4	0	50	50		X	X		
5	53.5	46.5	0		X			
6	46.5	53.5	0		X			
7	43	57	0		X			
8	53	40	7		X			
9	40	53	7		X			
10	40	60	0		X	X		
11	53	7	40		X	X		

**Table 2 materials-14-07223-t002:** Properties of the selected materials.

Property	1	2	3	4	5	6	7	9	11
Melting Temperature (°C)	333 [16] 336 [17] 330 [16]	308 [17] 310 [5]	279.8 [18] 271.0 [36]	220 [5]	221 [38] 222 [37] 227 [39]	222 [37]	137 [38] 145 [37]	220.9 [38] 220 [37]	142 [37] 138 [38]
Melting Enthalpy (kJ/kg)	266 [18] 116 [17] 266 [5]	174 [17] 199 [18] 172 [5]	199.5 [17]	100.7 [5]	99–110 [38] 100 [37] 108 [40]	117 [14]	98.6 [38] 97 [41]	142.3 [38] 142 [43]	80 [37] 17.2 [38]
Specific heat (kJ/kg K)	1.22 [5] 1.29 [42]	1.82 [5] 1.44 [42]	1.78 [42]	1.35 [5]	1.215 [38] 1.1 [41] 1.3 [4]	-	1.2 [38]	1.33 [38]	1.34 [38]

**Table 3 materials-14-07223-t003:** DSC results of the analyzed samples.

Sample Number	Mass Percentage of the Mixtures			DSC Parameters			
	KNO_3_ (wt %)	NaNO_3_ (wt %)	NaNO_2_ (wt %)	T onset (°C)	T end (°C)	Fusion Temperature (°C)	Enthalpy (J/g)
1	100	0	0	333.4	338.3	335.8	96.5
2	0	100	0	305.4	310.9	308.1	166.4
3	0	0	100	278.3	285.6	281.9	220.7
4	0	50	50	228.4	238.6	233.5	185.6
5	53.5	46.5	0	221.1	226.1	224.1	91.6
6	46.5	53.5	0	221.5	235.1	228.3	114.2
7	43	57	0	221.3	239.8	230.5	114.8
8	53	40	7	176.5	205.5	191	95.1
9	40	53	7	180.6	219.6	200.1	90.2
10	40	60	0	220.9	253.7	237.3	142.3
11	53	7	40	137.9	148.3	143.1	117.2

**Table 4 materials-14-07223-t004:** Specific heat of the evaluated compounds by DSC.

Compound	Mass	Molecular Weight	Heat Flux S–S (mW)	Heat Flux S–L	Cps		Cpl	
	(mg)	(kg/mol)		(mW)	(J/kg K)	(J/mol K)	(J/kg K)	(J/mol K)
1	6.5	0.10	0.55	0.11	507.69	51.33	99.69	10.08
2	6.9	0.08	1.52	0.85	1321.74	112.32	739.13	62.81
3	5.7	0.07	1.76	1.52	1848.42	127.50	1600.00	110.37
4	6.4	0.08	1.74	1.98	1629.38	125.43	1855.31	142.82
5	6.4	0.09	1.28	1.51	1198.13	112.15	1419.38	132.86
6	7.0	0.09	1.80	1.44	1542.86	142.68	1234.29	114.14
7	6.7	0.09	1.33	1.14	1191.04	109.47	1020.90	93.83
8	7.0	0.09	3.03	8.88	2597.14	239.99	7611.43	703.34
9	7.0	0.09	1.75	1.93	1499.14	135.39	1656.00	149.55

**Table 5 materials-14-07223-t005:** Results of the TGA analysis.

Parameters	Sample		
	(A) KNO_3_	(B) NaNO_3_	(C) NaNO_2_
Initial weight of the sample (mg)	3.810	3.600	3.740
Weight loss (mg) at 250 °C	0.018	0.025	0.056
Percentage of weight loss (%) at 250 °C	0.47	0.70	1.51
Weight loss (mg) at 600 °C	0.169	0.188	0.064
Percentage of weight loss (%) a 600 °C	4.41	5.25	1.72

## Data Availability

Data presented in this study are openly available in https://repositorio.uisek.edu.ec/.

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
