# Peer review of "Thermal Storage of Nitrate Salts as Phase Change Materials (PCMs)"

_materials, 2021, doi:10.3390/ma14237223_

Round 1
Reviewer 1 Report
In the present study, the energy storage potential of nitrate salts for renewable applications is investigated. Authors prepared 11 samples of nitrate salts as phase change materials (PCM) with sodium nitrate (NaNO3), sodium nitrite (NaNO2), and potassium nitrate (KNO3) as base materials. The results are new and of practical interest in thermal energy storage applications. The manuscript can be accepted after addressing the attached minor comments:

Author Response
Dear reviewer
We are very grateful to you for your review of our article. The time you have taken to give us reviews improves our original article.
Next, we are going to respond to the reviews that you formulate us
1- Consider name and unit for axis of all figures including Figs. 2-4.
In relation to this question, an edition of the title of the figures has been made to clarify the name of the absrobance. All reviews have been highlighted in yellow.
2- Please add the thermal conductivity data if available. This would help the analysis of the samples in engineering designs.
We cannot address this review because we do not have thermal conductivity equipment in the laboratories used.
3- Include the fusion temperatures and latent heat in table 4 or a new table. The data of thermal conductivity also can be added to this table or table 4.
The melting temperature has been included in table number 3. The latent heat data appear in table 4 for the solid and liquid states. Unfortunately we do not have equipment for thermal conductivity. All reviews have been highlighted in yellow.
4- The phase change nano / micro particles can also be suspended in a liquid for heat transfer and energy storage applications. Please update the literature review this point. For example, see: (a) Forced convection heat transfer of Nano-Encapsulated Phase Change Material (NEPCM) suspension in a mini-channel heatsink; (b) Exergy and energy analysis of wavy tubes photovoltaic-thermal systems using microencapsulated PCM nano-slurry coolant fluid; (c) Time periodic natural convection heat transfer in a nano-encapsulated phase-change suspension. (d) Consecutive charging and discharging of a PCM-based plate heat exchanger with zigzag configuration.
We appreciate this comment, these investigations have been introduced in the last paragraph of the second page of the article. All reviews have been highlighted in yellow.
5- The quality of Fig. 6 is too low and it seems blurry. Please use a better-quality image if possible.
Unfortunately it is the image that the laboratory equipment gives us.
6- Please correct the caption and format of Fig. 7. The caption is jumped.
Image correction has been performed.
7- In the SEM images, add some graphics and specify the particles and other important characteristics.
In relation to this section, some comments have been added to the SEM results. These are the results that we achieve with a computer that is now not in operation.
Reviewer 2 Report
This work reports the characterization results of molten salts for thermal energy storage material. Recently there were many reports about nitrate mixtures as a high temperature since it’s high potential. In the reviewer’s opinion, since this study does not have novelty compared with existing study and there are many ambiguous points, this work is not enough to publish in a high major impact journal as Materials with following reasons.
- There are many existing studies about nitrate mixture as a solar salt. What are the improved points (novelty) of this study compared to the existing study? The reviewer could not feel any novelty.
- Specific information for materials that are used in this study (Grain size, Purity, Maker, density etc.). Because this study investigated thermal characteristics of materials, the author should mention the information of raw material and experimental conditions as much as possible.
- Discussion on the FT-IR, SEM, and TGA results is not enough. Just explaining the experimental results without detailed purpose of experiments and scientific discussion is not enough for the journal paper.
- The reason for the different melting point between values in this study and references.
- Sub-/super-script for units and chemical formulas.
- Unit for enthalpy change on Abstract.
Author Response
Dear reviewer
We are very grateful to you for your review of our article. The time you have taken to give us reviews improves our original article.
Next, we are going to respond to the reviews that you formulate us
- There are many existing studies about nitrate mixture as a solar salt. What are the improved points (novelty) of this study compared to the existing study? The reviewer could not feel any novelty.
This research work determines the thermo-physical properties of nitrate salts (NaNO3, NaNO2, and KNO3) and their binary and ternary combinations due to the discrepancy and errors associate with different experimental setups conducted in different studies. In this sense, this study tries to overcome the limitations in previous literature.
This idea is highlighted in yellow in the last paragraph of the introduction
- Specific information for materials that are used in this study (Grain size, Purity, Maker, density etc.). Because this study investigated thermal characteristics of materials, the author should mention the information of raw material and experimental conditions as much as possible.
The first paragraph of the materials has been edited and a reference has been made to each of the materials used in the study.
- Discussion on the FT-IR, SEM, and TGA results is not enough. Just explaining the experimental results without detailed purpose of experiments and scientific discussion is not enough for the journal paper.
Each of the results have been edited to answer this question. All reviews have been highlighted in yellow.
- The reason for the different melting point between values ​​in this study and references.
The melting of the salt in the DSC crucible is not uniform, for this reason there is a range of melting of the material. Furthermore, we are in this study, using binary and ternary mixtures, in some cases not tested. When melting occurs, the melting point may vary with respect to other data in the literature. In addition, the melting temperatures of the references are usually found around the start and end temperatures of the melt that we present in the study. Lastly, they may vary a bit because the samples were 99% pure.
This is the idea on which the authors rely
- Sub- / super-script for units and chemical formulas.
The entire text has been edited to correct this issue. All reviews have been highlighted in yellow.
- Unit for enthalpy change on Abstract.
The sections of the abstract and the conclusions have been edited to correct this issue. All reviews have been highlighted in yellow.
Reviewer 3 Report
In this work, "Thermal storage of nitrate salts as phase change materials (PCMs)", the authors studied the energy storage potential of nitrate salts for particular applications in energy systems that use renewable resources. To this end, thermal, chemical and morphological characterization of 11 samples of nitrate salts was conducted. Based on the extracted results, the authors claimed that these materials are suitable for medium- and high-temperature thermal energy storage systems. Overall, this manuscript has a strong potential for a second review after applying the issues and addressing the shortcomings listed below:
1-The authors should polish/revise some grammatical mistakes and typos along the manuscript. I invite the authors to read their manuscript carefully and make the required changes where necessary.
2-In the Introduction section, while discussing recent developments in the use of PCMs, the following work should be considered and cited to give a more general view to the possible readers of the work: [(i) Extracting the temperature distribution on a phase-change memory cell during crystallization, Journal of Applied Physics 120, 164504 (2016)].
3- Try to increase the resolution of each Figure (especially Figure 6).
4- For Figures 2-4: Try to increase the font size of the axes and the size of blue-colored texts. Plus, what is the meaning of ‘A’ on the y-axis? You can write like this ‘A(… (put the meaning here))’.
5- Try to increase the font size of the text inside in Figure 6.
6- It seems there is problem with the caption of Figure 7.
Author Response
Dear reviewer
We are very grateful to you for your review of our article. The time you have taken to give us reviews improves our original article.
Next, we are going to respond to the reviews that you formulate us
1-The authors should polish/revise some grammatical mistakes and typos along the manuscript. I invite the authors to read their manuscript carefully and make the required changes where necessary.
The entire text has been edited to correct this issue. All reviews have been highlighted in yellow.
2-In the Introduction section, while discussing recent developments in the use of PCMs, the following work should be considered and cited to give a more general view to the possible readers of the work: [(i) Extracting the temperature distribution on a phase-change memory cell during crystallization, Journal of Applied Physics 120, 164504 (2016)].
We appreciate this comment, this investigation has been introduced in the first paragraph of the second page of the article. All reviews have been highlighted in yellow.
3- Try to increase the resolution of each Figure (especially Figure 6).
The image has been enlarged so that the figure can be better read.
4- For Figures 2-4: Try to increase the font size of the axes and the size of blue-colored texts. Plus, what is the meaning of ‘A’ on the y-axis? You can write like this ‘A(… (put the meaning here))’.
In relation to this question, an edition of the title of the figures has been made to clarify the name of the absrobance. All reviews have been highlighted in yellow.
5- Try to increase the font size of the text inside in Figure 6.
Unfortunately it is the image that the laboratory equipment gives us.
6- It seems there is problem with the caption of Figure 7.
Image correction has been performed.
Round 2
Reviewer 2 Report
The author tries to reflect the reviewer's opinion and improve the manuscript.
Reviewer agreed to publish this paper on Materials.
Reviewer 3 Report
In its current form, the revised manuscript is suitable for publication.